# A Base Model Selection Methodology for Efficient Fine-Tuning

## Abstract

While the accuracy of image classification achieves significant improvement with deep Convolutional Neural Networks (CNN), training a deep CNN is a time-consuming task because it requires a large amount of labeled data and takes a long time to converge even with high performance computing resources. Fine-tuning, one of the transfer learning methods, is effective in decreasing time and the amount of data necessary for CNN training. It is known that fine-tuning can be performed efficiently if the source and the target tasks have high relativity. However, the technique to evaluate the relativity or *transferability* of trained models quantitatively from their parameters has not been established. In this paper, we propose and evaluate several metrics to estimate the transferability of pre-trained CNN models for a given target task by featuremaps of the last convolutional layer. We found that some of the proposed metrics are good predictors of fine-tuned accuracy, but their effectiveness depends on the structure of the network. Therefore, we also propose to combine two metrics to get a generally applicable indicator. The experimental results reveal that one of the combined metrics is well correlated with fine-tuned accuracy in a variety of network structures and our method has a good potential to reduce the burden of CNN training.

## 1 Introduction

As convolutional neural networks (CNNs) have been widely used in many applications, a CNN model tends to become deeper and larger which necessitates a large amount of training data. This makes the number of arithmetic operations required for training huge (Canziani et al., 2016), resulting in long training time and large energy consumption. Moreover, there are large demands to process CNNs in edge devices where training a CNN model is a challenging task due to there limited computing resources. Having an efficient CNN learning methodology for embedded systems is very important. Hence, techniques to reduce the computational complexity of CNN training with maintaining inference accuracy have been studied intensively so far. These include, for example, quantization (Moons et al., 2016) and pruning (Molchanov et al., 2017).

Since obtaining enough training data is usually difficult and expensive in real situations, there is a great demand to create an efficient CNN training methodology. This is the motivation of using transfer learning which optimizes a trained model for a target domain by transferring information from a related source domain (Caruana, 1997). Fine-tuning, one of the transfer learning techniques, is a promising approach to reduce the burden of CNN training.

In fine-tuning, a source CNN model trained with a large labeled dataset (source task) is adapted to a different application (target task) to create a new model (target model). For example, ImageNet (Russakovsky et al., 2015) is becoming a de facto dataset as a source task for image recognition target tasks. In fine-tuning, it is a common practice to truncate the last fully connected layer (FC layer) of the pre-trained network, replace it with a new one that is relevant to the target task, and freeze the weights of the other layers of the pre-trained network.

It is known that fine-tuning can be performed efficiently if the source and the target tasks have high relativity (Caruana, 1997). If we choose a suitable source model for a given target task from multiple source task candidates, time and energy consumption for training are likely to be reduced. There has not been enough research effort to evaluate whether the transferred parameters are suitable for another task so far. In this paper, we investigate several metrics to evaluate "transferability" of source

CNN models for certain target tasks quantitatively without actually performing a time-consuming fine-tuning process. The evaluation results reveal that some of the metrics are well correlated with fine-tuned accuracy and our technique will greatly reduce the burden of CNN training.

## 2  RELATED WORK

Kornblith et al. (2018) examined the fine-tuning effect on using ImageNet 1K classification task as a source task for various target tasks. They evaluated the correlation between the accuracy of the source task (ImageNet) and fine-tuned target tasks with 13 classification models on 12 image classification tasks. A positive correlation was observed and the correlation coefficient was very high on average ($r^2 = 0.86$). Based on the results, they claimed that ImageNet accuracy is a good predictor of fine-tuning performance. In their research, the focus is on the fine-tuned performance after target tasks are actually trained. Compare to that, we consider fine-tuned accuracy without actually performing a time-consuming training phase. Moreover, they did not apply it for small datasets such as Stanford Cars and FGVC aircraft which do not necessarily have high fine-tuned performance even using ImageNet as a source task, meaning a model pre-trained with a large data set does not always give high fine-tuned accuracy. This implies that selecting an appropriate source task for a given target task is very important.

In Mahajan et al. (2018), they proposed a method of pre-training with large amounts of images on Instagram using its hashtag as a label. They found that some of the design choices that are made in current network architectures are too tailored to ImageNet-1k classification. They also mentioned that pre-training with a large amount of weakly labeled data improves classification task, but they did not apply it for detection or segmentation tasks.

Sato et al. (2018) raised a problem of negative transfer in the medical imaging area. They found that the results of transfer learning sometimes become worse rather than learning from scratch. This may also suggest that having appropriate pre-trained source tasks are necessary to train better models with transfer learning.

Transferring the data in source domain to target domain is called "instance transfer". In Liu et al. (2018), they proposed a new technique to transfer knowledge safely by measuring relativity between the source task and the target task.

Bau et al. (2017) quantified interpretability of latent representations of trained CNNs by evaluating their featuremaps with a method similar to the segmentation. It is judged by whether featuremaps can identify a certain concept and the number of identified concepts indicates interpretability of the target model. In this research, the authors revealed that CNNs identify higher-level concepts such as objects at higher-level convolutional layers and lower-level concepts such as textures at lower-level convolutional layers.

Morcos et al. (2018) reported that a highly selective neuron or featuremap which responds to input data in a certain class is not important for higher accuracy. Instead, neurons or featuremaps activated by inputs of a variety of classes are more important. They pointed out the grandmother cell hypothesis in the neural network may not be correct.

Hinton et al. (2015) proposed a knowledge distillation approach which compresses the knowledge of a large computational expensive model (teacher model) to create a smaller neural network model (student model). The idea of the knowledge distillation is to train a student model with a transfer set provided by the teacher model. It is becoming a popular technique to compress neural net models. The base idea of transferring knowledge of larger models to a smaller model to get a computationally efficient model is originally proposed by Buciluă et al. (2006). Furlanello et al. (2018) has reported that repeating knowledge distillation leads to better generalization ability of trained models. In addition, Zhang et al. (2018) has proposed a new learning method with distilling the knowledge of multiple models mutually to get higher performance models.

In many previous research efforts suggested that there exists an optimal model architecture for a target task and a pre-trained model using a source task greatly affects the fine-tuned accuracy of the network in transfer learning. To the best of our knowledge, this is the first attempt to evaluate transferability of pre-trained models without actually performing fine-tuning. This is one of the

novel points in this work. Besides it, our technique can be used to truncate a CNN model as presented in Appendix A.2. The method could be another approach to compress CNN models.

## 3 MODEL SELECTION METRICS

In this paper, we propose the metrics to evaluate the transferability of trained CNNs for a certain target task without actually retraining the networks to accelerate a fine-tuning process. We calculate the metrics based on the outputs of the last convolutional layer (featuremaps) for each architecture when images of the target task are input to the CNN. We calculate metrics according to the following three hypotheses and in total six metrics are considered.

H1. If models' featuremaps are easy to be classified by fully connected layers, they are highly transferable for a given task.

H2. If models' featuremaps have more divergent information, they are highly transferable for a given task.

H3. If models' featuremaps are sparse, they are tolerant to over-fitting, thus they are preferable as source models for a given task.

Based on the first hypothesis, we propose three metrics which are described in subsection 3.1 to 3.3. Another metric described in subsection 3.4 is based on the second hypothesis and the other two metrics shown in subsection 3.5 and 3.6 are based on the third hypothesis.

The notation used here is listed in Table 1. For the explanation, we use a model architecture of AlexNet as an example. In that case, "the last convolutional layer" means Conv5 layer and the dimension of it is $\mathbf{F}_i = (256, 13, 13)$. Only a small number of images for a class is enough to compute some metrics and we use randomly chosen ten images in this paper.

The summary of the metrics described in the following subsections is presented in Table 2.

Table 1: Notation used in this paper

| Notation | Description |
|---|---|
| $\mathcal{D}$ | Target task dataset |
| $\mathbf{x}_i$ | $i$-th image in dataset $\mathcal{D}$ |
| $C_j$ | $j$-th class in dataset $\mathcal{D}$ |
| $N$ | Num. of all images in dataset $\mathcal{D}$ |
| $K$ | Num. of class in $\mathcal{D}$ |
| $n_j$ | Num. of images included in the $j$-th class |
| $\mathbf{F}_i$ | Tensor of featuremap for $i$-th image at the last conv. layer |
| $\bar{\mathbf{F}}$ | Average of all $\mathbf{F}_i$ in dataset $\mathcal{D}$ i.e. $\bar{\mathbf{F}} = \frac{1}{N}\sum \mathbf{F}_i$ |
| $\bar{\mathbf{F}}_{\text{class}\_j}$ | Average of all $\mathbf{F}_i$ in class $C_j$ i.e. $\bar{\mathbf{F}}_{\text{class}\_j} = \frac{1}{n_j}\sum_{x_i \in C_j} \mathbf{F}_i$ |

Table 2: Summary of the proposed metrics

| | Description | Better for | Hypothesis based |
|---|---|---|---|
| $S_1$ | Inter-class discreteness | Larger | H1,H2 |
| $S_2$ | Intra-class discreteness | Smaller | H1 |
| $S_3$ | Pseudo F value | Larger | H1,H2 |
| $S_4$ | Sum of absolute values of channel correlations | Smaller | H2 |
| $S_5$ | Featuremap sparsity | Smaller | H3 |
| $S_6$ | Featuremap steepness | Smaller | H3 |

### 3.1 METRIC BASED ON SUM OF VARIANCE OF CLASS AVERAGE

If featuremaps for different class images differ from each other, the images may be easily classified by the FC layers. Based on this consideration, we propose the following metric $S_1$ based on the sum of variance of class average of featuremaps.

$$S_1 = \sum_{k=0}^{K-1} d\left(\bar{\mathbf{F}} - \bar{\mathbf{F}}_{\text{class}\_k}\right)^2 n_k \tag{1}$$

This indicates the inter-class discreteness. The larger the $S_1$, the better it is.

### 3.2 METRIC BASED ON SUM OF CLASS-WISE FEATUREMAP VARIANCE

If featuremaps for different images in a class are similar, the images may be easily classified by the FC layers. Based on this consideration, we propose the following metric $S_2$ based on the sum of class-wise variance of featuremap in a class.

$$S_2 = \sum_{k=0}^{K-1} \sum_{x_i \in C_k} d\left(\bar{\mathbf{F}}_{\text{class}\_k-} - \mathbf{F}_i\right)^2 \tag{2}$$

This indicates the intra-class discreteness and is usually used as an objective function of k-means method. The smaller the $S_2$, the better it is.

### 3.3 METRIC BASED ON PSEUDO-F VALUE

The Pseudo-F is known as one of the metrics of clustering algorithms(Caliski & Harabasz, 1974). It is defined as the ratio of inter-class variance to intra-class variance of featuremaps defined by the following formula 3:

$$S_3 = \frac{S_1/(K-1)}{S_2/(N-K)} \tag{3}$$

The larger the $S_3$, the better it is.

### 3.4 METRIC BASED ON CORRELATIONS BETWEEN EACH FEATUREMAP CHANNEL OF AN IMAGE

Convolutional layers have multiple filters with different parameters and extract different features from input images. For example, the Conv5 layer of AlexNet outputs featuremaps with $13 \times 13 \times 256$ dimension extracted by 256 different filters. Wu et al. (2018) proposed a method to estimate contents of featuremaps by using correlations between channels of the featuremaps. Based on the previous observation, we propose the metric $S_4$ based on the absolute value of correlations between channels of featuremaps of each image. The $S_4$ is mathematically defined as follows.

$$S_4 = \sum_n \sum_{i,j} |\text{cor}\left(\mathbf{F}_n\left(i, :, :\right), \mathbf{F}_n\left(j, :, :\right)\right)| \tag{4}$$

Here, cor(A,B) indicates the Pearson's product moment correlation coefficient between the arguments of A and B. The smaller the $S_4$ value, the better it is.

### 3.5 METRIC BASED ON SPARSITY OF FEATUREMAPS

Yaguchi et al. (2018) stated that using Adam(Kingma & Ba, 2014) as the optimizer for training CNNs is an effective way to suppress over-fitting and observed that weights of convolutional layers become sparse as a result of Adam. They also mentioned that the sparsity of featuremaps after a ReLu layer become more sparse as training progresses. Based on this observation, we propose the metric $S_5$ which indicates the sparsity of featuremaps as follows:

$$S_5 = \frac{1}{N} \sum_i \left(1 - \frac{(\text{Num. of zero elements in } \mathbf{F}_i)}{(\text{Num. of elements in } \mathbf{F}_i)}\right) \tag{5}$$

The smaller the $S_5$ value, the better it is.

### 3.6 METRIC BASED ON STEEPNESS OF FEATUREMAPS

Based on the third hypothesis, it is expected that if there are a large number of zero elements in featuremaps and only a few elements have larger values, the network well captures the feature of the images. We then propose the metric $S_6$ which quantifies the *steepness* of the featuremaps as follows:

$$S_6 = \frac{1}{N} \sum_i \frac{(\text{Num. of high value elements in } \mathbf{F}_i)}{(\text{Num. of zero elements in } \mathbf{F}_i)} \tag{6}$$

The number of high value element in $\mathbf{F}_i$ means that the number of elements in featuremaps whose difference from the maximum element value is less than 1. The smaller the $S_6$ value, the better it is.

Table 3: Configurations for pre-training and fine-tuning.

| Parameter | Description |
|---|---|
| epoch | 30 |
| Loss function | Cross-entropy |
| Optimizer | Stochastic gradient descent (SGD) |
| Learning rate | 0.01 (beginning) and 0.1x every 7-epoch |

## 4 EXPERIMENTAL SETUP

Throughout the experiments, we evaluate the relation between the fine-tuning accuracy and the proposed metrics for each base model. The evaluation flow is as follow.

1. Prepare base models trained with various source tasks and several network architectures

2. Input the images of target training sets to each base model

3. Get the featuremap from the last convolutional layer of each base model

4. Perform fine-tuning for each target task using all the based models

5. Evaluate the correlation between fine-tuned accuracy and proposed metrics

On the first step of the flow, we use various ImageNet subsets for preparing the base models as described in subsection 4.1. The target tasks to be evaluated is described in 4.2. On the fifth step, we use the Spearman's rank correlation coefficient (Spearman's $\rho$) as a criterion of the effectiveness of the metrics. We explain the details of it in Appendix A.1.

### 4.1 PREPARING PRE-TRAINED CNN MODELS

We prepared CNN models trained with different source tasks for evaluating our method. We use various subsets of ImageNet1K 2012 training set as source tasks. A total of 60 subsets of ImageNet are created with two policies, "class-wise division" and "horizontal division".

In "class-wise division", the image classes in the ImageNet dataset are divided into several sets each with 100 classes. For example, classes from class No. 1 to No. 100 form a subtask. Similarly, classes from class No. 101 to No. 200 form another subtask. We also create smaller subtasks with 50 classes.

As for "horizontal division", the total number of classes in each subtask is the same as ImageNet but the images of each class are divided to form subtasks. The original ImageNet dataset has 1000 classes each with around 1300 images. If we divide it into ten subtasks, each subtask has 1000 classes each of which has about 130 images. We use two cases of 10 and 20 divisions in this policy.

As model architectures, AlexNet, VGG16, and ResNet18 are used to prepare the base models. We pre-train these architectures using the subtasks mentioned above to prepare the base model. The configurations of pre-training are summarized in Table 3. Note that this configuration is also used for the fine-tuning phase.

### 4.2 TARGET TASKS

Table 4 shows the datasets used as the target tasks in the experiment. We evaluate fine-tuned accuracy with eight image classification datasets whose training set size differs ranging from 2,040 to 8,144 images. The pre-trained base models are fine-tuned to each target task independently. We measure the top-1 accuracy for all the tasks.

Table 4: Datasets examined in this research

| Dataset | Classes | Size(train/test) |
|---|---|---|
| Stanford Cars (Krause et al., 2013) | 196 | 8,144/8,041 |
| FGVC Aircraft (Maji et al., 2013) | 100 | 6,667/3,333 |
| Describable Textures (DTD) (Cimpoi et al., 2014) | 47 | 3,760/1,880 |
| Oxford-IIIT Pets (Parkhi et al., 2012) | 37 | 3,680/3,369 |
| Caltech-101 (Fei-Fei et al., 2004) | 102 | 3,060/6,084 |
| Oxford 102 Flowers (Nilsback & Zisserman, 2008) | 102 | 2,040/6,149 |

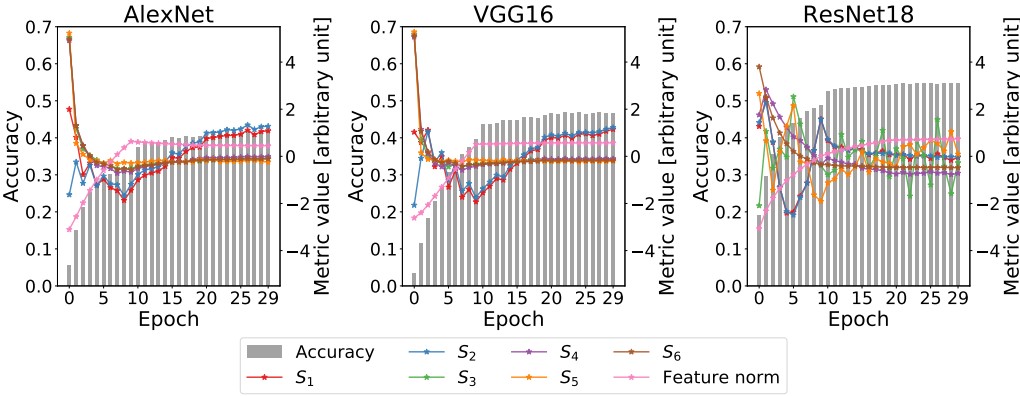

Figure 1: Validation accuracy vs. proposed metrics on normal ImageNet training.

## 5 RESULTS

### 5.1 THE RELATIONSHIP BETWEEN ACCURACY AND METRICS

We first investigate how the proposed metrics are related to the inference accuracy within the normal training phase. We trained the ImageNet 1K classification task from scratch and recorded both the proposed metrics and the validation accuracy in each epoch. The metrics were calculated with featuremaps of the last convolutional layer for each model architecture.

Figure 1 shows the validation accuracy (on the left Y-axis) and proposed metric values (on the right Y-axis) for each epoch (X-axis) in ImageNet training for three network architectures. All the metrics are normalized so that the mean is 0 and the variance 1.

For the metrics of $S_1$ and $S_3$, the higher the better. On the other hand, the smaller the better in the other metrics. Since the results for AlexNet and VGG16 have a similar trend and are easy to analyze, we discuss those results first. As training proceeds, the validate accuracy improves and all the metrics except for $S_1$ and $S_2$ go into small value. $S_1$ and $S_2$ have irregular trends at the beginning of training, but become high as validation accuracy improves. These indicate that the proposed metrics and the inference accuracy have some correlations. Though we expected that $S_2$ and $S_3$ metrics would go into larger and smaller as training proceeds, the results were opposite. It seems that the CNNs prefer diversified vectors as the featuremap results for higher classification accuracy even when intra-class images are applied.

As for the case of ResNet18, there seem to have some trends but they are not very clear. This may come from the effect of normalization layers. However, $S_4$ and $S_6$ have relatively clear and similar trends with the cases of AlexNet and VGG16. In general, these metrics are expected to be appropriate metrics for predicting the suitability of the convolutional layers for given tasks.

We also show the feature norm value of the last convolutional layer in Figure 1. We found that S1 and S2 do not have any correlation with the feature norm, but S6 does show a negative correlation.

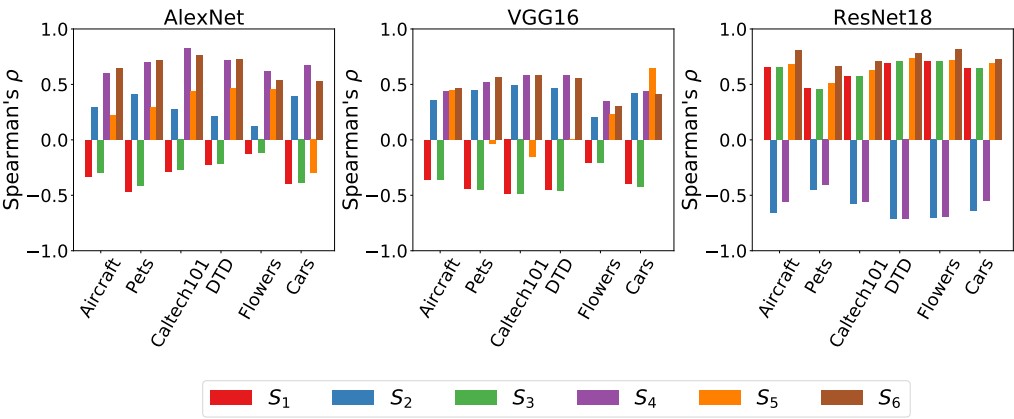

Figure 2: Spearman's rank correlation between fine-tuning accuracy and the proposed metrics.

## 5.2 MODEL SELECTION EFFICIENCY

We next evaluate the ability of proposed metrics to select a suitable model for a given target task from different pre-trained models. The results for several target tasks (in X-axis) are summarized with Spearman's $\rho$ (ranking correlations among 60 pre-trained models) in Figure 2. Higher absolute values of Spearman's $\rho$ indicate good correlations, showing the effectiveness of the metrics.

Overall, $S_4$ and $S_6$ have a high correlation in all the network architectures. If the architecture is the same, there in no significant difference in given tasks, but the trend of each metric is different for different architectures. Specially, the results in ResNet18 show different trends compared with the other two architectures. For example, $S_4$ shows a relatively high positive correlation for AlexNet and VGG16 while it has a high negative correlation for ResNet18. We conjecture that this is because ResNet18 has a unique neural network structure among the three, that is, batch normalization layers. Each model architecture extracts image features in different ways and the metrics reflect those difference appropriately. Though most of the metrics have high correlations, all of them cannot be used for selecting suitable base models in a general way. However, $S_6$ shows consistently high positive correlation in many cases and could be one of the most suitable metrics.

## 5.3 COMBINING TWO METRICS

In the previous subsection, we show that not every single metric can be a generally good indicator of selecting suitable base models. In this subsection, we combine any two metrics using the following weighted sum to create new indices.

$$C_{i,j}(\alpha) = (1 - \alpha)S_i + \alpha S_j \qquad (7)$$

where $\alpha \in [0, 1]$, $i = 1, ..., 5$ and $j = i + 1, ..., 6$.

To combine two metrics fairly, we normalize all the metrics considering all the network architectures and target tasks so that each of them is within the range of -1 to 1. In addition to that, we invert $S_2$, $S_4$, $S_5$, and $S_6$ so that all the metrics exhibit better correlation for positive values.

Figure 3 presents the results of all the combinations of $C_{i,j}$. The horizontal and vertical axes of each graph show $\alpha$ and the Spearman's rank correlation between fine-tuning accuracy and values of $C_{i,j}(\alpha)$, respectively. The value of $\alpha$ is changed from 0 to 1 with the increment of 0.001 in all the experiments.

Again the trends for AlexNet (solid lines) are similar to VGG16 (dotted lines). In these cases, the combinations of $S_4$ and $S_5$, $S_4$ and $S_6$, or $S_5$ and $S_6$ give relatively high correlation independent from the given $\alpha$. In the case of ResNet18 (dashed line), any combinations which include $S_5$ or $S_6$ have high correlation if we apply an appropriate $\alpha$. For all the three architectures, $C_{5,6}$ exhibits fairly high correlations with fine-tuned accuracy. Both $S_5$ and $S_6$ are based on the sparsity of

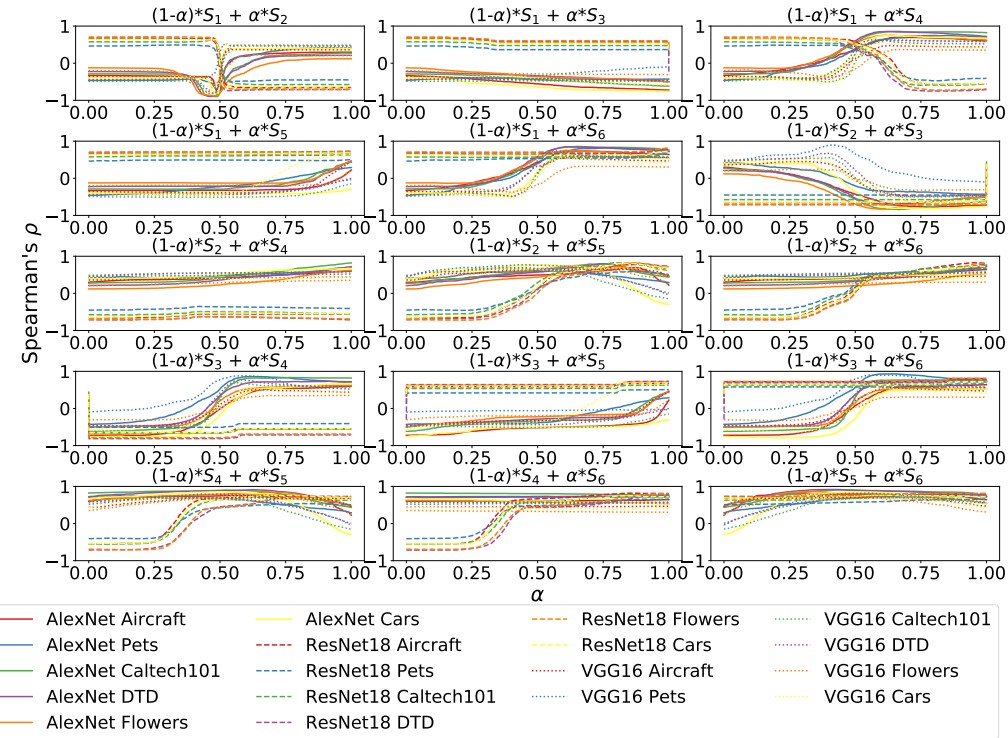

Figure 3: The rank correlations for the combinations of two metrics.

the featuremaps, hence it is suggested that the sparsity is a specially important factor to evaluate transferability of pre-trained models.

Focusing on the combination of $S_5$ and $S_6$, $C_{5,6}(0.574) = 0.426 S_5 + 0.574 S_6$ gives the highest rank correlation for all combinations of architectures and target tasks on average. The average rank correlation between fine-tuning accuracy and $C_{5,6}(0.574)$ is 0.7514 and the variance is 0.005993. The maximum and minimum $\rho$ are 0.8874 and 0.5854 obtained in *Caltech101* with AlexNet and *Pets* with ResNet18. Since it is known that high correlation is recognized if $\rho$ is larger than about 0.7, we can conclude that $C_{5,6}$ is well correlated to the fine-tuning accuracy and can be used for selecting some suitable base models without actually performing time-consuming training. Note that $S_5$ and $S_6$ metrics can be calculated without target task labels, hence $C_{5,6}$ can be easily calculated which would reduce the burden of the fine-tuning process further.

## 6 CONCLUSION

In this paper, we proposed a method to evaluate the transferability of convolutional layers of trained CNN models for a certain target task. Our experiments demonstrated that some of the proposed metrics can be a good indicator of fine-tuned model accuracy. We also showed that combining two metrics, especially the sparsity and the steepness of featuremaps, gives us a more efficient model selection metric. This result may suggest that the proposed method with some metrics is useful in selecting suitable base models among a number of candidate networks.

In this paper, we focus on a base model selection methodology, but the proposed technique is easily applicable to an appropriate layer selection methodology. Using all the convolutional layers in a base model does not always provide a good classification accuracy after fine-tuning. We can shrink the network size by truncating the convolutional layers and the suitable final convolutional layer can be selected by using the proposed metric. The details are described in Appendix A.2.

One of the clear extensions of this work is to evaluate the proposed technique with other network model architectures and other image recognition tasks such as object detection and semantic seg-

mentation. It is reported that pre-training with a large number of images is not effective in object detection or semantic segmentation (Mahajan et al., 2018) and there is no clear relation between ImageNet classification accuracy and the image recommendation ability (del Rio et al., 2018). Studying transferability for those tasks is also our future work.

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

## A APPENDIX

### A.1 RANK CORRELATIONS BETWEEN METRICS AND TRANSFER ACCURACY

Spearman's rank correlation coefficient or Spearman's $\rho$ is a non-parametric measure of statistical dependence between the rankings of two variables (rank correlation). It is denoted as follows:

$$\rho = 1 - \frac{6 \sum d_i^2}{M(M^2 - 1)} \tag{8}$$

where $d_i$ means the difference between the two ranks of each value and $M$ is the number of values. We evaluate this value between the rank of each proposed metric and one of fine-tuning accuracy.

In general, it is said that the order of two values have a higher correlation if $|\rho| \geq 0.7$.

### A.2 SELECTING A SUITABLE LAYER FOR RUNCATING CNN MODELS

It is well known that the lower layers of a CNN usually recognize low-level local features such as edges, corners, spots in a generic fashion. It progresses toward higher layers to identify higher-level features which are optimized for a certain task. There is a possibility that the lower convolutional layers extract features which is better suited for a target task in fine-tuning compared with the last conv. layer. We next evaluate the effect of proposed metrics for selecting the most transferable layer in a given pre-trained model.

### A.2.1 CONCEPT

In order to evaluate the layer selecting capability with the proposed metrics, we use a fine-tuning method shown below. We use only a lower part of the CNN model and prune the deeper part. Proper pooling and FC layer are added to the selected lower layers. We call this "truncate fine-tuning".

### A.2.2 RESULTS

In the experiment, we use only ResNet18 as a base model architecture. Figure 4 shows the accuracy of truncate fine-tuning when varying the selected convolutional layer. The base model is trained for 30 epochs with ImageNet dataset. The horizontal axis indicates the index of the convolutional layer which is used for the last conv. layer and the vertical axis means fine-tuning accuracy for each target task.

Overall, the results are better when using all the conv. layers. However, truncating one or two layers does not have much impact on accuracy. For example, in case of the Flowers dataset, the accuracy is 0.92 when using 16th conv. layer, while it is 0.91 and 0.87 when using 15th and 14th conv. layer, respectively. Moreover, in case of Caltech101, the accuracy of the 7th layer is better than that of the 8th layer.

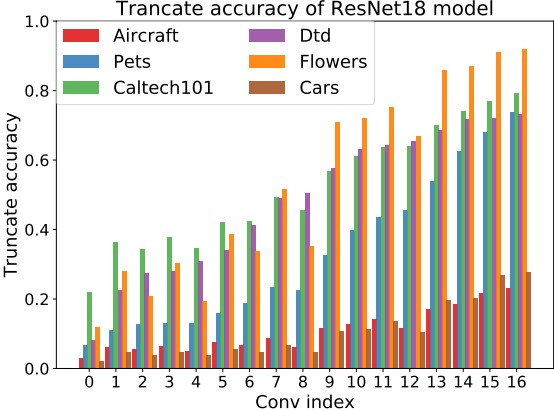

Figure 4: Truncate fine-tuning accuracy for each selected convolutional layer in ResNet18

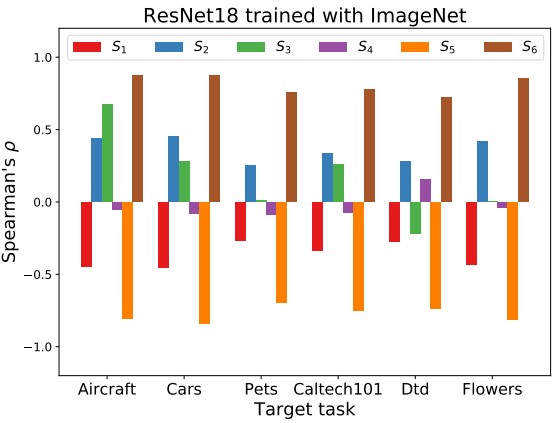

Figure 5: Rank correlations between the proposed metrics and truncate fine-tuning accuracy

We can use the proposed metrics to select the suitable convolutional layer for efficient truncate fine-tuning. Figure 5 shows the Spearman's rank correlation between truncate fine-tuning accuracy and the proposed metrics. For all the target tasks, Spearman's $\rho$ between $S_6$ metric and truncate fine-tuning accuracy is larger than 0.7. This result shows that the $S_6$ metric is useful for selecting

an appropriate layer for truncate fine-tuning and the proposed method has a potential to effectively compress the base CNN model in fine-tuning process.

