# OpenReview forum: "A Base Model Selection Methodology for Efficient Fine-Tuning"
_ICLR.cc/2020/Conference — Reject_

### Official Review · AnonReviewer2 · 2019-10-12
**Official Blind Review #2**

**Rating:** 3

**Review:**

In this paper, the authors tried several metrics, from which they pinpoint one as the indicator to select a pretrained model without practically fine-tuning. I appreciate that the authors addressed an important problem, while the experimental setup and results are less convincing. Therefore, the proposed metric(s) and settings in the current shape are not that practical.

Pros:
-	In this work, the authors focused on an important problem – selecting the best pretrained model from a zoo of models without finetuning all of them in a brute-force manner.
-	The idea of applying the metric(s) to select a layer from which the consecutive layers are truncated is intriguing, for the sake of model compression.

Cons:
-	The work is more alike a course project than a novel scientific contribution, with all the metrics mainly inherited from the evaluation of clustering and other existing literatures.
-	The experiments are not comprehensive, so that the conclusions drawn are weak and untenable.
-	The writing with many grammatical errors and typos definitely needs polishing.

Detailed comments:
1.	The authors calculated all the metrics in terms of feature maps in the last layer. Therefore, it is not valid to conclude that other metrics are less effective than sparsity. Some metrics, like fisher discriminator, are likely effective only in the low-level features. It is better for the authors to investigate all metrics in terms of all layers.
2.	Since all the pretrained models are pretrained from ImageNet, it is possible that they lose some diversity, which tends to deactivate metrics other than S5 and S6. Unless the authors trained several models from scratch on different datasets and achieved similar results, the conclusion that S5 and S6 are strong is not that convincing.
3.	Why during fine-tuning, do you use SGD instead of Adam?
4.	What kind of correlation metrics do you use in Eqn. (4)? And will the correlation metric influence the effectiveness of S4?
5.	The most confusing part lies in that the valid metric suggested by the author, i.e., C5,6(0.574), is only an empirical observation on a very limited number of datasets, without any principled methodology. Actually, it is highly possible that given a highly different dataset, the best metric changes. Should the practitioners determine which metric to use first, and even the coefficient to combine metrics?  This is paradox, in my opinion.
6.	Grammatical errors and missing details:
o	Abstract: a variety of network structure -> a variety of network structures
o	Section 3: The sentences for the three hypotheses, H1/2/3, do not even have verbs in the if part.
o	Section 3.4: to estimates -> to estimate
o	Section 3.6: which quantify -> which quantifies
o	…

**Experience Assessment:**

I have published in this field for several years.

**Review Assessment: Checking Correctness Of Derivations And Theory:**

I carefully checked the derivations and theory.

**Review Assessment: Checking Correctness Of Experiments:**

I carefully checked the experiments.

**Review Assessment: Thoroughness In Paper Reading:**

I read the paper thoroughly.

---

> ### Author Response · Authors · 2019-11-15
> **Response to review #2**
>
> 1. Why during fine-tuning, do you use SGD instead of Adam?
> Reply: Since SGD is simpler than Adam, we thought that the results and findings using SGD could be more general and interpretable.
>
> 2. What kind of correlation metrics do you use in Eqn. (4)? And will the correlation metric influence the effectiveness of S4?
> Reply: Thanks for asking. We use Pearson's product moment correlation between each channel of the featuremaps. We have made this clear in the revised paper. Since Pearson's product moment correlation is a very general metric, we think there is not any bias for the effectiveness of S4.

---

### Official Review · AnonReviewer3 · 2019-10-19
**Official Blind Review #3**

**Rating:** 3

**Review:**

This paper aims to speed up finetuning of pretrained deep image classification networks by predicting the success rate of the process without running it. The authors suggest running samples from the target task on the (trained) source model, and computing a few sensible measures from the final output layer which indicate how well the trained features are separating the target images. Many experiments are presented, and some of them seem promising.

Overall the idea is, while simple, interesting and potentially promising: DL Training costs have been rising significantly in the past few years, and being able to make sensible predictions about which source dataset/task combination best fits a target task would be a great contribution. Nonetheless, there are some methodological concerns that cast doubt about the presented results, which would need to be addressed before making this paper ready for publication.

Comments:

1. The authors present multiple experimental results, many of which indicate a somewhat noisy signal. The only method that works on most tasks, combining S5 and S6, is somewhat underreported. How many different \alpha values did the authors consider? how were they selected? Was the same alpha values used for each all experiments? Given this large set of measures and combinations of measures experimented with in this paper, I am left wondering whether this approach would generalize to new target datasets.

2. A major assumption the authors are making is that there is a single number that determines whether a given trained architecture would transfer well to a new target task. But the finetuning process is affected by many factors (e.g, hyperparameters, random seeds), and thus it might be that with a different hyperparameter selection, the observed correlations would look completely different. I would have liked to see at the very least an analysis of the correlation between multiple runs of finetuning that show that they are well correlated before computing correlations with external measures.

3. The authors point out that transfer learning works poorly on medical imaging (Sato et al., 2018). It would be interesting to experiment with such datasets and observe whether the proposed method is able to make accurate predictions in this domain.

4. The experiments regarding truncating CNNs seem interesting, and I was disappointed to find out they are not part of the main paper.

Other comments:

1. Moons et al. (2016) and Molchanov et al. (2017) speed up *inference* and not training.

2. The authors argue that Mahajan et al. (2018) "did not apply it (their method) for detection or segmentation tasks.". However, neither did this paper.

3. Writing:
-- Several typos and grammatical errors across the paper. For instance:
- "*In* many previous research efforts suggested" ("in" should be dropped)
- Hypothesis H1 is ungrammatical

-- Many of the citations were in the wrong format (intro: Canziani et al. 2016), repetitive (3.5: Yaguchi et al.), etc.



**Experience Assessment:**

I have published one or two papers in this area.

**Review Assessment: Checking Correctness Of Derivations And Theory:**

N/A

**Review Assessment: Checking Correctness Of Experiments:**

I assessed the sensibility of the experiments.

**Review Assessment: Thoroughness In Paper Reading:**

I read the paper thoroughly.

---

> ### Author Response · Authors · 2019-11-15
> **Response to review #3**
>
> 1. How many different $\alpha$ values did the authors consider in the combined metrics? Were the same alpha values used for each experiment?
> Can this approach be generalized to new target datasets?
>
> Reply: The value of $\alpha$ is changed from 0 to 1 with an increment of 0.001 in all the experiments. We have made this clear in the revised paper.
> For all model architectures and datasets, we searched for a combination of two metrics by changing the alpha that maximizes the average rank correlation coefficients.
> Since we evaluated several cases and found the optimal combination, we believe this approach can be applied even for new datasets.

---

### Official Review · AnonReviewer1 · 2019-10-23
**Official Blind Review #1**

**Rating:** 3

**Review:**

The authors propose several metrics to evaluate the transferability of pretrained CNN models for a target task without actually fine-tuning the networks to accelerate fine-tuning.

The paper has done some interesting empirical studies on how to predict the transferability of a neural network. The motivation of performing model selection without actual finetuning has many benefits for practical applications and I believe this is the right direction. In this perspective, the paper is novel. However, my major concern is a lack of in-depth analysis and thorough verification of the proposed metrics.

First, there is no clear relationship between the six evaluation metrics and the fine-tuning performance. Figure 1 depicts some correlation, however, it is not clear enough. The difference in ResNet18 is not further investigated. Given the dominant usage of ResNets and its variations in practice, it is important to provide analysis for the reasons behind.

Second, some of the proposed metrics (S1/S2) are actually closely related with weight norms.  The definition of high value elements in S6 is also based on the absolute difference with the maximum feature map values. Does the author apply weight decay during training? Normally the weight norm will becomes smaller and so could the featuremap values. The S1/S2/S6 metrics could therefore be influenced. I would like to see figures showing the relationship between the weight/feature norms and the metric during training.

Third, it turns out that the only useful metrics are the combination of feature sparsity and feature steepness (i.e., C_56). The authors should make it more specific and clear for their contributions. Why only two combination of the metrics is considered? Does the best selected coefficient generalize to other models and datasets?

Finally, according to https://arxiv.org/abs/1805.08974, better pretrained ImageNet model also generalize better. I would expect a deeper ResNets should outperform AlexNet and VGG16.  What is the relationship between the proposed metrics and the ImageNet performance?

Minor:

The S3 metric is actually “a ratio of inter-class variance to intra-class variance” rather than “ ratio of intra-class variance to inter-class variance”.


**Experience Assessment:**

I have read many papers in this area.

**Review Assessment: Checking Correctness Of Derivations And Theory:**

I assessed the sensibility of the derivations and theory.

**Review Assessment: Checking Correctness Of Experiments:**

I carefully checked the experiments.

**Review Assessment: Thoroughness In Paper Reading:**

I read the paper at least twice and used my best judgement in assessing the paper.

---

> ### Author Response · Authors · 2019-11-15
> **Response to review #1**
>
> 1. During training, does the author apply weight decay by which some of the metrics could be affected? The reviewer would like to see figures showing the relationship between the weight/feature norms and the metric during training.
>
> Reply: We would like to thank the reviewer for pointing this out and valuable suggestions. We did not actually use weight decay during training. Based on the reviewer's suggestion, we have added feature norms for the last convolutional layer into Figure 1 of the revised paper. We found that S1 and S2 do not have any correlation with the feature norm, but S6 does show a negative correlation.

---

### Author Response · Authors · 2019-11-15
**General response**

First of all, we would like to express our gratitude to all reviewers for their thoughtful reviews and valuable comments.  We attempted to reply to some of the questions and comments in the individual reply field for each reviewer.

---

### Decision · Program_Chairs · 2019-12-19

**Decision:**

Reject

**Comment:**

This paper proposes to speed up finetuning of pretrained deep image classification networks by predicting the success rate of a zoom of pre-trained  networks without completely running them on the test set. The idea is that a sensible measure from the output layer might well correlate with the performance of the network. All reviewers consider this is an important problem and a good direction to make the effort. However, various concerns are raised and all reviewers unanimously rate weak reject. The major concerns include the unclear relationship between the metrics and the fine-tuning performance, non- comprehensive experiments, poor writing quality. The authors respond to Reviewers’ concerns but did not change the major concerns. The ACs concur the concerns and the paper can not be accepted at its current state.